# Diabetic Neuropathic Pain and Serotonin: What Is New in the Last 15 Years?

**DOI:** 10.3390/biomedicines11071924

**Published:** 2023-07-06

**Authors:** Nazarine Mokhtar, Stephane Doly, Christine Courteix

**Affiliations:** NEURO-DOL, INSERM (Institut National de la Santé et de la Recherche Médicale), Université Clermont Auvergne, 63000 Clermont-Ferrand, France; nazarine.mokhtar@uca.fr (N.M.); stephane.doly@uca.fr (S.D.)

**Keywords:** serotonin, painful diabetic neuropathy, animal models

## Abstract

The neurotransmitter serotonin (5-hydroxytryptamine, 5-HT) is involved in numerous physiological functions and plays a key role in pain modulation including neuropathic pain. Diabetic neuropathy is a common complication of diabetes mellitus often accompanied by chronic neuropathic pain. Animal models of diabetes offer relevant tools for studying the pathophysiological mechanisms and pharmacological sensitivity of diabetic neuropathic pain and for identifying new therapeutic targets. In this review, we report data from preclinical work published over the last 15 years on the analgesic activity of drugs acting on the serotonergic system, such as serotonin and noradrenaline reuptake inhibitor (SNRI) antidepressants, and on the involvement of certain serotonin receptors-in particular 5-HT_1A_, 5-HT_2A/2c_ and 5-HT_6_ receptors-in rodent models of painful diabetic neuropathy.

## 1. Introduction

The aim of this review is to present recent advances and discoveries in preclinical research on painful diabetic neuropathy, focusing on serotonin (5-hydroxytryptamine, 5-HT) in the peripheral and central nervous systems involved in the modulation of pain.

Neuropathic pain is “pain caused by injury or disease affecting the somatosensory nervous system” [1]. It can be caused by a wide variety of peripheral or central disorders such as viral infection (HIV, zona), trauma or surgery, nerve compression or infiltration tumors, toxic substances (alcohol, chemotherapy agents), stroke or multiple sclerosis. Neuropathic pain has a significant impact on quality of life, cognitive performance [2,3] and is often associated with depression and anxiety [4]. A major provider of neuropathy and neuropathic pain is diabetes. Painful diabetic neuropathy is defined as “pain as a direct consequence of abnormalities in the peripheral somatosensory system in patients with diabetes” [5]. The prevalence of painful neuropathy in patients with type 1 and type 2 diabetes mellitus (T1D, T2D) is estimated at 14.7% and 24.7%, respectively [6], and considering the 537 million people with diabetes worldwide, it constitutes a major humanistic and economic burden [7]. Clinical manifestations of diabetic neuropathy are various and mainly include diabetic distal symmetric sensory or sensorimotor neuropathy developing as a “dying-back” neuropathy. It affects the most distal extremities first then extends to the trunk and occasionally to the upper limbs corresponding to a “glove and stocking” distribution [8]. Diabetic neuropathic pain is difficult to manage. Tight glycemic control is the primary strategy for preventing or slowing the progression of diabetes complications, including neuropathy. The symptomatic treatment of diabetic neuropathic pain is based on the prescription of tricyclic antidepressants (particularly amitriptyline), serotonin–noradrenaline reuptake inhibitors (SNRIs, duloxetine and venlafaxine) and the calcium channel alpha-2-delta ligand gabapentin. However, these treatments offer, at best, partial relief in only one out of three patients [9]. Selective serotonin reuptake inhibitor (SSRI) antidepressants (such as fluoxetine) are not effective enough to be recommended in neuropathic pain patients [9]. This clinical observation is discrepant with the control of pain processing exerted by spinal serotonergic projections originating from the brainstem [10]. In light of this, the role of the central serotonergic system (i.e., abnormal serotonergic neurotransmission) and the resistance of diabetic neuropathic pain to SSRI antidepressants have been investigated.

## 2. Experimental Models of Diabetes in Rodents

Animal models mirroring human diabetes are of great importance not only to study functional and structural abnormalities characterizing diabetic neuropathy but also to better understand pathophysiological mechanisms underlying neuropathic pain, identify new targets, and evaluate new therapeutic strategies. Rodents are currently used to reproduce experimental T1D and T2D either through dietary (high-fat diet-fed), chemical (streptozotocin, alloxan) or genetic manipulations (zucker diabetic fatty (ZDF) rat, obese leptin-deficient (ob/ob) mice, leptin receptor-deficient (db/db) mice, nonobese diabetic (NOD) mice) or spontaneous diabetes [11,12,13]. In the next paragraphs, we will present briefly experimental models of diabetes currently used.

## 3. Chemically-Induced Diabetic Animals

Streptozotocin (STZ) is by far the most commonly used agent to induce chemical pancreatectomy in rats and mice. STZ is an antibiotic extracted from streptomycin presenting an analogy with glucose which enters and accumulates in pancreatic β-cells via the glucose transporter type 2 (GLUT2) [14]. The cytotoxic effect of STZ is due to DNA alkylation and protein glycosylation. Diabetes is currently induced by either single or multiple intraperitoneal (i.p.) or intravenous (i.v.) injections of STZ depending on the animal species (rat or mouse), the route of administration (i.v. or i.p.), the age of the animals and the type of diabetes (T1D or T2D) desired [13,15,16]. The model obtained by a single large dose of STZ is more consistent with T1D, whereas a smaller STZ dose is more consistent with T2D. Rodents are frequently studied over 4–12 weeks of diabetes and, to ensure a good clinical condition during the entire experiment, diabetic animals may receive regular low-dose insulin injections. Only 2–4 weeks are needed to study features such as allodynia and slowing of nerve conduction velocity, whereas significantly longer durations (at least 8–16 weeks) are required to reveal some structural features of the neuropathy. Structural pathology of nerve trunks is limited to a reduction in axonal caliber with late (more than 4 months) myelin thinning, occasional segmental demyelination and fiber loss [16,17,18]. Other chemical compounds available to induce diabetes in rodents are alloxan, ferric nitrilotriacetate, or dithizone but rarely used [12,17,19].

## 4. Spontaneous Animal Models of T1D

The NOD mouse and BioBreeding (BB) Wistar rat are the two most commonly used animals that spontaneously develop diseases with similarities to human T1D. These animals have been inbred in laboratories for many generations to be selected for hyperglycemia [16,19].

NOD mice develop insulitis at about 3–4 weeks of age. At this pre-diabetic stage, the pancreatic islets are infiltrated by CD4+ and CD8+ lymphocytes, although B cells and NK cells are also present. Similarities in T1D genes between NOD mice and humans have been extremely useful in dissecting some mechanisms and pathways behind T1D. Thus, these mice are potentially suitable for testing therapies in which modulation of the autoimmune response is being targeted [13,19,20,21]. The Akita mouse, represents an alternative to NOD mice since diabetes, occurring at the age of 3–4 weeks, results from a spontaneous mutation of the insulin 2 gene leading to incorrect proinsulin processing, misfolded protein aggregation and pancreatic beta-cell death [12,19,21].

BB rats usually develop spontaneous autoimmune diabetes just after puberty with a similar incidence in males and females. Approximately 90% of rats develop diabetes between 8 and 16 weeks of age with the classical features of T1D such as hyperglycemia, hypoinsulinemia, and weight loss. The diabetic phenotype is quite severe and the rats require insulin therapy to survive. Although the animals have insulitis with the presence of T cells, B cells, macrophages and NK cells, the animals are lymphopenic with a significant reduction in CD4+ T cells and a near absence of CD8+ T cells. The model provided insight into the genetics of T1D [12,16,17,19].

## 5. Spontaneous Animal Models of T2D

Obese ob/ob and db/db mice are two models of spontaneous T2D resulting from leptin and leptin receptor deficiency, respectively. At the age of 3–4 weeks, ob/ob mice display the main features of T2D such as hyperglycemia, compensatory hyperinsulinemia, hyperlipidemia, polyphagia and excessive weight gain associated with lowered physical activity. Db/db mice spontaneously overexpress insulin and display hyperglycemia, hyperinsulinemia and insulin resistance and develop obesity [21,22]. Concerning rats, the most used spontaneous model of T2D is the Zucker Diabetic Fatty (ZDF) which results from the leptin receptor gene deficiency responsible for hyperglycemia, glucose intolerance, polyphagia, excess in fat storage and hypertension [12,17,21].

Recent advances in understanding the pathophysiology of neuropathic pain have highlighted the role of abnormal serotonergic neurotransmission and downstream signaling pathways in the development and maintenance of neuropathic pain. Serotonin is a peripheral and central pain neuromodulator and neurotransmitter that also plays a key role in various functions such as thermoregulation, sleep, mood, appetite, and psychiatric and brain disorders (anxiety, depression, schizophrenia, obsessive-compulsive disorders, Alzheimer’s, autism). As a modulator of pain, 5-HT is released from serotonergic neurons mainly issued from the rostroventral medulla (RVM) in the brainstem and projecting into the spinal cord (Figure 1). There, serotonin will activate serotonergic receptors localized on primary afferent fibers, nociceptive-thalamic projection neurons, local interneurons or descending projections including serotonergic ones. Depending on the subtype of 5-HT receptor targeted, serotonin will have a pronociceptive or antinociceptive effect.

## 6. Alteration of Serotonergic Neurotransmission in Diabetic Neuropathic Pain Models

We performed bibliographic research in MEDLINE-PubMed limited to 15 years (2008–2023) using the following keywords and combinations: “diabetes” and “serotonin” and “pain” and “rats” or “mice”. We found articles of preclinical studies reporting a role for serotonin/serotonin receptors in the pathophysiology of painful diabetic neuropathy, and others reporting the analgesic effect of drugs such as SNRI and SSRI antidepressants, melanocortin, or molecules (such as epicatechin) all sharing an effect on serotonin neurotransmission.

### 6.1. Effect of Compounds Acting on Serotonergic Pathways

The involvement of the endogenous serotonergic system in diabetic neuropathy can be explored in its entirety using drugs that reduce or increase central serotonin neurotransmission. Thus, para-chloroamphetamine (PCA), a neurotoxic agent involving an initial release of 5-HT from presynaptic nerve endings and subsequently leading to a degeneration of 5-HT axonal endings, is widely used in preclinical research investigating the contribution of serotonin in the pathophysiology of pain and pharmacology of painkillers. Accordingly, PCA administration was shown to reduce the anti-hyperalgesic effect (on mechanical hyperalgesia) of the repeated administration of venlafaxine (an SNRI) in STZ diabetic rats, suggesting a serotonergic mechanism in this effect [23]. The antidepressant also potentiated the antinociceptive action of morphine in the same context [24]. Pre-treatment clearly enhanced the analgesic activity of morphine, but the prolonged administration (21 days) of venlafaxine completely abolished the analgesic action of morphine in the STZ rat model [25]. Like venlafaxine, amitriptyline, a tricyclic antidepressant and duloxetine, an SNRI, are part of the first-line treatment of neuropathic pain in the French recommendations [26]. Combination therapy (antidepressant combined with gabapentinoids) proposed as second-line treatment [26] has been tested in the STZ rat model [27]. Using the tail flick test in order to assess thermal hyperalgesia, the authors found that the combination of duloxetine (20 mg/kg) or amitriptyline (10 mg/kg) with pregabalin (100 mg/kg) results in anti-hyperalgesia consistent with that obtained after the administration of each drug with a 50% reduction in the effective doses (40, 20 and 200 mg/kg, respectively). Duloxetine alone (10 mg/kg), also reduced mechanical allodynia and thermal hyperalgesia in STZ diabetic rats [28]. Rather than a reinforcement of serotonin modulation, the analgesic effect of duloxetine was due, in this study, to the reinforcement of noradrenergic signals in the spinal cord by inhibiting the reuptake of noradrenaline, shown to be impaired in painful diabetic neuropathy. Mixcoatl-zecuatl et al. [29] showed that i.p. and i.t., but not s.c. administration of duloxetine attenuated tactile hypersensitivity in STZ rats. In the murine model of STZ-induced diabetes, duloxetine (5, 10 and 20 mg/kg, i.p.) dose-dependently reduced thermal hyperalgesia [30]. As well as in alloxan diabetic mice, the antidepressant improved mechanical allodynia and thermal hyperalgesia [31]. In addition, duloxetine provided peripheral and central neuroprotective effects related to the downregulation of spinal astrocytes and microglia [31]. The reinforcement of noradrenaline descending inhibition induced by duloxetine was also sufficient to improve pain-related behavior in STZ diabetic rat [32] but without the curative-like effects observed with the isomer of indeloxazine, AS1069562. Indeed, AS1069562 showed persistent analgesia even after treatment discontinuation, which could be mediated by restoration of insulin-like growth factor 1 (IGF1) and fibroblast growth factor 2 expression levels, resulting in the amelioration of impaired nerve function. The importance of IGF1 has also been shown by Morgado et al. [33] in mediating functional plasticity in the brain under diabetic condition. Indeed, early treatment with IGF1 normalized serotonin and noradrenaline levels (increased in diabetes), but also improved mechanical hyperalgesia and reduced neuronal hyperactivity in the ventrolateral periaqueductal gray (PAG) and dorsal horn of the spinal cord. Other antidepressants, a SNRI, milnacipran, and two SSRIs, paroxetine and fluvoxamine, i.t. injected, induced an antiallodynic effect on neuropathic pain in STZ diabetic rats [34] whereas the SSRI had no effect on allodynia in rats with chronic constriction nerve injury, suggesting differential involvement of serotonin system depending on the etiology of the neuropathy [34]. Ammoxetine, another potent SNRI emerged as a long-lasting analgesic and anti-depressive therapeutic in the rat model of STZ-induced diabetic neuropathy. The extensive microglial activation observed 6 weeks after diabetes induction and the production of cytokines TNF-α and IL-1β, were reduced by repeated administration of the antidepressant [35]. Furthermore, inhibition of the hyper-phosphorylation of intracellular signaling molecules, such as p38 and JNK, was described as the transduction mechanism underlying the analgesic effect of ammoxetine in the model of STZ-induced diabetic rat [35].

Mianserin is a tetracyclic antidepressant with a high affinity for the 5-HT_2_ receptor, histaminergic H1 and α_2_-adrenergic receptors. Its effect on noradrenergic neurotransmission was examined in STZ diabetic rats. It reduced mechanical and thermal hyperalgesia/allodynia. These effects were reversed by α-methyl-para-tyrosine methyl ester (inhibitor of catecholamine synthesis), phentolamine (nonselective α-adrenergic receptor antagonist), propranolol (nonselective β-adrenergic receptor antagonist), and naloxone (opioid receptor antagonist), but not para-chlorophenylalanine methyl ester, a serotonin synthesis inhibitor. These results suggest that the beneficial effect of mianserin on diabetic neuropathic pain is mediated by an increase in catecholamine but not serotonin levels in the synaptic cleft as well as by interactions with both adrenoreceptors and opioid receptors but not 5-HT receptors [36].

1-Methyl-1,2,3,4-Tetrahydroisoquinoline (1MeTIQ) is an endogenous amine present in the human brain with neuroprotective, free radical scavenging, antidepressant, anticonvulsant, anti-addictive, anxiolytic and pro-cognitive properties. Acute administration of 1MeTIQ in STZ diabetic mice attenuated mechanical allodynia and thermal hyperalgesia. Ondansetron a nonselective antagonist of the ionotropic 5-HT_3_ receptor (permeable to incoming Na^+^ and Ca^2+^ ions), yohimbine (α_2_-adrenergic receptor antagonist) or naloxone were shown to reverse the anti-hyperalgesic effect induced by 1MeTIQ, involving serotonergic, noradrenergic and opioidergic mechanisms. In STZ diabetic mice, serotonin concentrations were reduced in the prefrontal cortex, striatum and hippocampus as were dopamine levels in the striatum while 1MeTIQ was able to restore serotonin and dopamine concentrations to normal levels [37].

Another way to explore the serotonin system is to target serotonin receptors with agonists, inverse agonists or antagonists. Serotonin receptors are metabotropic (G-protein-coupled receptors (GPCR)) and ionotropic (ligand-gated ion channel) receptors, classified in seven families and counting no less than 14 subtypes. Among these receptors, 5-HT_1A_, 5-HT_2A/2c_ and 5-HT_6_ receptor subtypes remain the most studied in diabetic neuropathic pain.

### 6.2. Effect of Compounds Acting on 5-HT_1A_ Receptors

There are multiple reports on the involvement of 5-HT_1A_ receptors in both the analgesic effects of dual antidepressants and 5-HT-acting drugs. The 5-HT_1A_ receptor is either pre- or post-synaptic and negatively coupled to adenylyl-cyclase (A.C.) activity through a Gi/o protein leading to a reduction in cyclic adenosine monophosphate (cAMP) and to opening K^+^ channels [38]. This results in the membrane hyperpolarization of neurons expressing this receptor. The neutral antagonist WAY100635 is currently used to explore the involvement of 5-HT_1A_ receptors in the analgesic effect of drugs. Thus, Jesus et al. [39] showed the involvement of 5-HT_1A_ receptors in the analgesic effect of cannabidiol (CBD) on mechanical allodynia associated with neuropathic pain in STZ diabetic rats. In their experiment, the intrathecal injection of WAY100635 completely prevented the anti-allodynic effect of CBD suggesting a role for descending serotonergic pathways and 5-HT_1A_ receptor in CBD effect. The 5-HT_1A_ receptor is also involved in the analgesic effect of epicatechin (a flavonoid present in cacao, green tea, grape seeds, strawberries, red wine) since the neutral 5-HT_1A_ antagonist WAY100635 prevented the anti-hyperalgesic effect of epicatechin on both phases (i.e., acute and inflammatory pain) of formalin-induced nociception in STZ diabetic rats [40]. In adult ZDF rats, repeated auricular electric stimulations which excite the parasympathetic nervous system, can elevate the circulating 5-HT concentration, upregulate the expression of central 5-HT_1A_ receptors and stop the development of nociceptive sensitivity [41]. NLX-112, a drug candidate intended for the treatment of L-DOPA-induced dyskinesia and a highly selective 5-HT_1A_ receptor full agonist was shown to reduce tactile allodynia in STZ-induced diabetic neuropathic mice by returning the von Frey hair threshold to almost normal levels [42].

### 6.3. Effect of Compounds Acting on 5-HT_2A/2C_ Receptors

5-HT_2A/2C_ receptors also belong to the superfamily of GPCR. They drive a depolarizing effect on neuronal membranes by increasing intracellular Ca^2+^ (which enhances spontaneous excitatory postsynaptic currents) and inositol phosphate concentrations secondary to their coupling to Gq/11 and activation of phospholipase (PL)C, PLA2 and the extracellular signal-regulated kinase (ERK) pathway [43]. Both pro-algesic and anti-hyperalgesic effects of 5-HT_2A_ receptor activation have been reported in experimental models of neuropathic pain (see review in Courteix et al., 2018) [44]. The intracellular post-synaptic-density (PSD)-95 protein, known to interact with the C-terminus of the 5-HT_2A_ receptor, is increased in the spinal cord neurons of STZ diabetic neuropathic rats [45]. Decoupling 5-HT_2A_ receptor/PSD-95 interaction with a peptidyl mimetic (TAT-2ASCV) inhibited mechanical hyperalgesia and suppressed spontaneous temperature preference behavior (i.e., thermal allodynia) induced by diabetic neuropathy. This effect was shown to be mediated by the activation of 5-HT_2A_ receptors by endogenous 5-HT since it was abolished by co-injection of the 5-HT_2A_ antagonist M100907. Therefore, it was concluded that the 5-HT_2A_ receptor/PSD-95 interaction contributes to the impairment of the 5-HT_2A_ receptor functionality. More importantly, TAT-2ASCV peptide injected together with fluoxetine strongly enhanced the antihyperalgesic action of fluoxetine in diabetic neuropathic rats, suggesting that SSRIs may be effective against neuropathic pain provided that spinal 5-HT_2A_ receptors are disconnected from their associated PSD-95 proteins. The same results were observed in carrageenan-induced inflammatory pain [46] and traumatic neuropathic pain [47]. In STZ diabetic rats, Bektas et al. [48] showed that the anti-hyperalgesic effect of the anti-epileptic drug zonisamide involves 5-HT_2A_ receptors since the neutral 5-HT_2A_ antagonist ketanserin partially and totally inhibited the enhancement of the mechanical and thermal thresholds, respectively. Intrathecal administration of ketanserin or pruvanserin (5-HT_2A_ receptor antagonists) also blocked the anti-allodynic effects of duloxetine, also suggesting the involvement of spinal 5-HT_2A_ receptors in the effect of antidepressants [29]. Agomelatine, a new class of antidepressants and acting as an MT1 and MT2 melatonergic receptor agonist was shown to produce anti-hyperalgesic effects in STZ diabetic rats by involving 5-HT_2C_ receptors [49].

### 6.4. Effect of Compounds Acting on 5-HT_6_ Receptors

5-HT_6_ receptors are positively coupled to A.C./protein kinase A (PKA) signaling, leading to the production of cAMP and neuronal depolarization. In addition to this canonical pathway, the 5-HT_6_ receptor was shown to engage the mechanistic target of rapamycin (mTOR) [50] and cyclin-dependent-kinase 5 (Cdk5) pathways [51]. 5-HT_6_ receptor, like other serotonin receptors, is also able to be spontaneously active independently of endogenous serotonin activation [38]. The initial demonstration of in vivo constitutive activity of 5-HT_6_ receptors in preclinical models of traumatic (spinal nerve ligation) and toxic (oxaliplatin injection) neuropathic pain was conducted with the demonstration of concomitant activation of mTOR signaling by constitutively active 5-HT_6_ receptors [52]. Recently, 5-HT_6_ receptor inverse agonists [53,54], including SB258585 (first described as a neutral antagonist [55], and also rapamycin (mTOR inhibitor), was shown to reverse mechanical hyperalgesia and reduce cognitive co-morbidities in STZ diabetic rats, demonstrating the role of the constitutive activity of 5-HT_6_ receptors in diabetic neuropathic pain and associated cognitive deficits and that of mTOR activation by constitutively active 5-HT_6_ receptors [56]. In this study, the intrathecal administration of a cell-penetrating mimetic peptide that disrupts the physical interaction between the 10 amino-acyl residues of the C-terminus of the 5-HT_6_ receptor and mTOR protein [52], also suppressed mechanical hyperalgesia in STZ diabetic rats [56]. The attenuation of thermal hyperalgesia in STZ diabetic mice after systemic administration of SB258585 and the absence of changes in spinal serotonin levels during the course of diabetes further support a role for 5-HT_6_ receptor constitutive activity in diabetic neuropathic pain [57]. In addition to 5-HT_6_ receptor-dependent mTOR activation, mTOR has been shown to be involved in glucose homeostasis by regulating pancreatic β-cell function [58] and neuronal function. In the PAG, mTOR phosphorylation as well as PI3K and Akt phosphorylation were shown to be increased in STZ diabetic rats [59]. Whereas the increase in pro-inflammatory cytokines concentrations is contemporaneous with PI3K/Akt/mTOR activation, cytokine receptor blockade reduced the amounts of p-mTOR and p-PI3K in the PAG and decreased mechanical allodynia and thermal hyperalgesia in diabetic rats. This highlights alterations in descending pain inhibitory projections from the PAG in the pathophysiology of diabetic neuropathic pain. In dorsal spinal cord, mTOR activation can also be due to adaptor protein APPL1 deficiency [60]. Indeed, APPL1 was found to be drastically reduced in dorsal spinal cord of STZ diabetic rats and the APPL1 overexpression was able to impair the mTOR activation and alleviate mechanical hyperalgesia in diabetic rats [60] as did rapamycin injection [61]. In fact, APPL1 regulates negatively the activity of mTOR via a positive and negative modulation of AMPK and Akt, respectively [60]. Finally, inhibition of the mTOR signaling pathway and inhibition of phosphorylation of the proteins downstream of mTOR, S6K1 and 4E-BP1 in sensory nerves, is one of the mechanisms by which exercise intervention ameliorated mechanical hypersensitivity in STZ diabetic rats [62].

### 6.5. Comparison of Animal Studies Exploring the Serotonergic System in Diabetic Neuropathic Pain Models

Several points can be highlighted from these preclinical studies, compiled in Table 1. Firstly, all the compounds tested in these articles have been shown to be effective in treating neuropathic pain and its comorbidities. Then, most of the studies were carried out in males, and only three [29,40,48] included females (Table 1) whereas women show a similar prevalence of diabetic neuropathic pain as men. As recently reviewed [63], mechanisms of pain hypersensitivity after nerve injury involving neuron-immune-glia interactions may or not differ according to sex. Indeed, microglia appears to play a major role in sex differences in neuropathic pain, macrophages appear as sex-independent mediators of neuropathic pain and astrocytes, are mediators of pain hypersensitivity in both sexes [63]. In the present review, only one study performed in both males and females [48] reports no sex difference. In female rats, diabetes was reported to induce painful neuropathy characterized by tactile hypersensitivity and chemical hyperalgesia to formalin injection [28,39].

Another point, with regard to STZ-induced diabetes, is the different modalities of diabetes induction. For example, various routes of administration of STZ were used, either intramuscularly [23,24,25], intravenously [28,32,34,36,48] or intraperitoneally [27,29,33,35,39,40,45,49,56,59,60,61,62]. These different methods do not allow us to compare data in an effective way, and even if some studies used identical routes of injection, the dose is different, ranging from 45 mg/kg [59] to 75 mg/kg [27]. These modalities together with different diabetes durations (comprised between 1 to 10 weeks) [29,31,33,60,62] can induce different diabetes severity and associated diabetic complications, including neuropathy and painful behaviors. In experimental models of diabetes, the greater the hyperglycemia is, the more the state of general health is altered, with a consequent risk of bias in pain behavior. This is why, to avoid this bias and for evident ethical reasons, it has been recommended to treat diabetic rats with a low dose of long-lasting insulin (2 IU every other day) and keep diabetic animals alive for no longer than 4 weeks following STZ injection [64]. However, the introduction of insulin treatment has only been reported in three studies [45,49,56].

Finally, the majority of animal studies used mechanical devices to assess tactile allodynia (von Frey hair test) or mechanical hyperalgesia (paw pressure test) and few of them used a thermal device (paw immersion, tail flick, hot/cold plate, Hargreaves tests) to assess cold allodynia or hot hyperalgesia. In diabetic patients, the characteristics of neuropathic pain assessed using the DN4-interview questionnaire (a simple questionnaire designed to diagnose neuropathic pain in four questions) are very different from those assessed in animals; they include numbness (around 65% of patients), tingling (65%), burning (55–60%), pins and needles (55–60%) and electric discharges (around 50%); less frequent are pain due to cold (25–30% of diabetic patients) and itching (around 25% of diabetic patients) [6]. Altered quality of life, comorbid symptoms of anxiety, sleep disturbance, depression and cognitive deficits are more prevalent in patients with chronic pain with neuropathic characteristics than those with chronic pain without neuropathic characteristics [6]. Despite this, only two studies included the assessment of diabetes-related comorbidities [35,56].

## 7. Conclusions

We report data from the scientific literature obtained in experimental models of painful diabetic neuropathy with high translational value since the pharmacology is close to that observed in humans and the clinical symptoms of neuropathic pain are robust and reproducible. Most behavioral studies carried out to assess neuropathic pain rely on the application of acute nociceptive (to assess hyperalgesia) or non-nociceptive (to assess allodynia) stimuli to elicit a painful response. Spontaneous pain behavior was not evaluated. The majority of studies was performed in male rats whereas diabetes and painful diabetic neuropathy affect in equal proportion men and women. While we focused on diabetic neuropathic pain, chemotherapy-induced peripheral neuropathy and trauma-induced neuropathic pain models often but not systematically share identical pathophysiological mechanisms [65]. The alteration in monoaminergic neurotransmission and, in particular, serotonergic transmission, may explain the ineffectiveness of SSRIs in neuropathic pain patients [9] and the alteration of receptor functionality may explain some pathophysiological mechanisms of painful neuropathy. The review also highlights the need to assess the multidimensional aspects of pain, which are all too often neglected in preclinical studies, even though they can be a means of assessing quality of life, which is now widely evaluated in clinical trials. Finally, focusing on certain serotonin receptors such as 5-HT_6_ receptors shown to be constitutively active in this particular context, and their signaling pathways, will probably lead to the identification of a specific target and/or transduction mechanism which, modulated by pharmacological agents, will provide effective and safe analgesics. With the number of diabetic patients expected to reach 626 million by 2045 [11], this is a new challenge that can only be successfully met with the collaboration of preclinical and clinical researchers working together in translational research for the benefit of the patient.

## Figures and Tables

**Figure 1 biomedicines-11-01924-f001:**
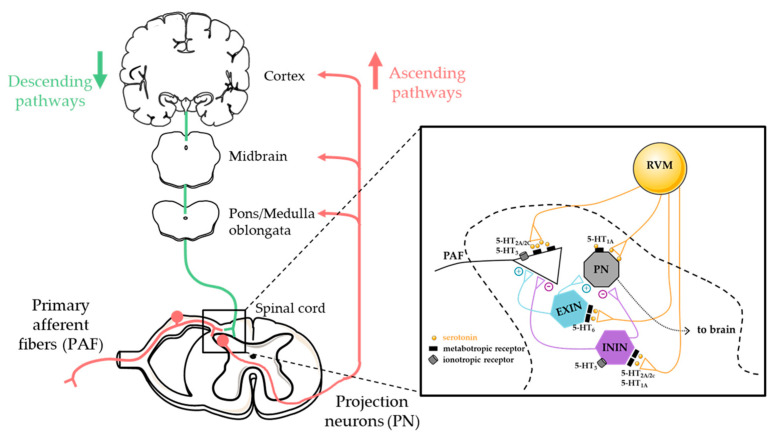
Descending serotonin pathways and 5-HT receptors in the dorsal horn spinal cord. RVM: Rostroventral medulla, EXIN: excitatory interneuron, ININ: inhibitory interneuron, PN: Projection neuron.

**Table 1 biomedicines-11-01924-t001:** Animal studies of diabetic neuropathic pain exploring the serotonergic system published between 2008 and 2023.

	Model	Diabetes Duration	Sex	Sensory-Discriminative Aspects		Associated Co-Morbidities (Test)	Treatment Used to Improve Pain and Associated Comorbidity
				Tactile Allodynia/Mechanical Hyperalgesia (Test)	Thermal Allodynia/Hyperalgesia (Test) or Chemical Hyperalgesia		
Tawfik et al. 2018 [31]	Alloxan mice (180 mg/kg, i.p.)	10 weeks	M	+(von Frey hair)	+(Hot plate)	n.e.	Duloxetine (SNRI)
Li et al. 2018 [41]	Zucker diabetic fatty	8 weeks	M	+(von Frey hair)	+(Hot, Hargreaves)	n.e.	Repeated auricular electric stimulation
Sari et al. 2019 [57]	STZ Mice (150 mg/kg, i.p.)	5, 9–15 days	M		+(Hot plate)	n.e.	SB258585 (5-HT_6_ receptor inverse agonist)
Tokhi et al. 2023 [37]	STZ Mice (200 mg/kg, i.p.)	4 weeks	M	+(von Frey hair)	+(Hot tail immersion) +(Tail flick)	n.e.	1-Methyl-1,2,3,4-Tetrahydroisoquinoline (endogenous amine)
Sałat et al. 2017 [42]	STZ Mice (200 mg/kg, i.p.)	3 weeks	M	+(von Frey hair)	+(Hot plate)	n.e.	NLX-112 (selective 5-HT_1A_ receptor agonist)
Kuhad et al. 2009 [30]	STZ Mice (200 mg/kg, i.p.)	4 weeks	M		+(Hot tail immersion) +(Hot plate)	n.e.	Duloxetine (SNRI)
Cegielska-Perun et al. 2012 [25]	STZ Rats (40 mg/kg, i.m.)	19–39 days	M	+(Paw pressure) +(von Frey hair)		n.e.	Venlafaxine (SNRI)
Cegielska-Perun et al. 2013 [23]	STZ Rats (40 mg/kg, i.m.)	19–39 days	M	+(Paw pressure) +(von Frey hair)		n.e.	Venlafaxine (SNRI)
Cegielska -Perun et al. 2015 [24]	STZ Rats (40 mg/kg, i.m.)	19–39 days	M	+(Paw pressure) +(von Frey hair)		n.e.	Venlafaxine (SNRI)
Murai et al. 2014 [32]	STZ Rats (45 mg/kg, i.v.)	3–7 weeks	M	+(Electronic von Frey)		n.e.	Isomer of Indeloxazine (Antidepressant)
Ikeda et al. 2009 [34]	STZ Rats (50 mg/kg, i.v.)	2–4 weeks	M	+(von Frey hair)		n.e.	Milnacipran (SNRI) Paroxetine, Fluvoxamine (SSRI)
Bektas et al. 2014 [48]	STZ Rats (50 mg/kg, i.v.)	3–8 weeks	M F	+(Paw pressure)	+(Hot plate) +(Hot tail immersion)	n.e.	Zonisamide (Antiepileptic)
Üçel et al. 2015 [36]	STZ Rats (50 mg/kg, i.v.)	4 weeks	M	+(Tail clip) +(Paw pressure) +(Dynamic plantar)	+(Hot plate) +(Hot, Hargreaves) +(Cold plate)	n.e.	Mianserin (Tetracyclic antidepressant)
Kinoshita et al. 2013 [28]	STZ Rats (50 mg/kg, i.v.)	6 weeks	M	+(von Frey hair)	+(Hot, Hargreaves)	n.e.	Duloxetine (SNRI)
Mixcoatl-Zecuatl et al. 2011 [29]	STZ Rats (50 mg/kg, i.p.)	6–10 weeks	F	+(von Frey hair)		n.e.	Duloxetine (SNRI)
Quinonez-Bastidas et al. 2013 [40]	STZ Rats (50 mg/kg, i.p.)	2 weeks	F		+(Formalin test)	n.e.	Epicatechin (Flavonoid)
Guo et al. 2021 [59]	STZ Rats (45–60 mg/kg, i.p.)	3–5 weeks	M	+(von Frey hair)		n.e.	Cytokine receptor blockade (reducing p-PI3K and p-mTOR)
Zhang et al. 2018 [35]	STZ Rats (60 mg/kg, i.p.)	4 weeks	M	+(von Frey hair)		+(Open Field)	Ammoxetine (SNRI)
Jesus et al. 2019 [39]	STZ Rats (60 mg/kg, i.p.)	4 weeks	M	+(von Frey hair)		n.e.	Cannabidiol
He et al. 2019 [60]	STZ Rats (60 mg/kg, i.p.)	1–4 weeks	M	+(von Frey hair)	+(Cold tail immersion)	n.e.	APPL1 overexpression (Inhibiting mTOR)
He et al. 2016 [61]	STZ Rats (60 mg/kg, i.p.)	3 weeks	M	+(von Frey hair)		n.e.	Rapamycin (Inhibiting mTOR)
Ma et al. 2020 [62]	STZ Rats (60 mg/kg, i.p.)	1–5 weeks	M	+(von Frey hair)		n.e.	Exercise intervention (Inhibiting mTOR)
Morgado et al. 2011 [33]	STZ Rats (60 mg/kg, i.p.)	1–4 weeks	M	+(Paw pressure)		n.e.	Insulin-like growth factor 1
Chenaf et al. 2017 [49]	STZ Rats (72 mg/kg, i.p.)	3 weeks	M	+(Paw pressure)		n.e.	Agomelatine (Antidepressant)
Pichon et al. 2010 [45]	STZ Rats (72 mg/kg, i.p.)	3 weeks	M	+(Paw pressure)	+(Plate preference)	n.e.	TAT-2ASCV (Uncoupling of 5-HT_2A_ receptor/ PDZ protein)
Mokhtar et al. 2023 [56]	STZ Rats (75 mg/kg, i.p.)	3 weeks	M	+(Paw pressure)		+(NOR)	PZ1388, 1Z1386, PZ1179, SB258585 (5-HT_6_ receptor inverse agonists) TAT-VEPE (Uncoupling of 5-HT_6_ receptor/ mTOR) Rapamycin (Inhibiting mTOR)
Tripathi et al. 2016 [27]	STZ Rats (75 mg/kg, i.p.)	2–6 weeks	M		+(Tail flick)	n.e.	Duloxetine (SNRI) + Amitripyline (Tetracyclic antidepressant) or + Pregabalin (Gabapentinoid)

Abbreviations and symbols: F, female; i.m., intramuscular; i.p., intraperitoneal; i.v., intravenous; M, male; mTOR, mechanistic target of rapamycin; n.e., not evaluated; NOR, novel object recognition test; PI3K, phosphatidyl inositol 3 kinase; SNRI, serotonin and noradrenaline reuptake inhibitor; SSRI, selective serotonin reuptake inhibitor; STZ, streptozotocin; +, observed.

## Data Availability

Not applicable.

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
