# Peer review of "Diabetic Neuropathic Pain and Serotonin: What Is New in the Last 15 Years?"

_biomedicines, 2023, doi:10.3390/biomedicines11071924_

Round 1

Reviewer 1 Report

Mokhtar and coworkers review the analgesic effects of antidepressants such as 5-HT- and noradrenaline-reuptake inhibitors (particularly, SNRI) and drugs acting on 5-HT receptors (particularly, 5-HT1A, 5-HT2A/2C and 5-HT6) in diabetic neuropathy rodent models produced by STZ application and so on.  This review article appears to be based on the papers published between 2008-2023.  Table 1 summarizes the rodent models mentioned in this article.  There are sentences including terms that are used without explanation or are not scientifically correct here and there.  Thus, there are many points that should be addressed and may serve to amend this manuscript, as follows:

Major points:

1.     Table 1 demonstrates diabetic neuropathy rodent models including diabetes duration, sex and hyperalgesia type.  Although the text states what kind of drug the hyperalgesia is alleviated by, this fact is not shown in Table 1.  For example, it is stated in lines 157-159 that thermal hyperalgesia in the murine model of STZ-induced diabetes is reduced by duloxetine [30], but this result is not given in Table 1.  Please amend this point.  If no drug is shown to relieve hyperalgesia, Table 1 does not appear to be of value.

2.     The authors state data obtained using not only SNRI but also SSRI, tricyclic and tetracyclic antidepressants.  A list of antidepressants should be provided for the convenience of the general reader who is unfamiliar with the drugs.

3.     Lines 224 and 225: “.. submitted to the formalin test” is difficult to understand, because formalin produces not only acute but also inflammatory pain.  Please amend this point.

4.     Lines 233 and 234: it is not clear why sEPSC increase produces a membrane depolarization by increasing intracellular Ca2+.  Please make this point clear.

5.     Since various drugs related to 5-HT receptors appear in this review article, a list of 5-HT receptor drugs should be provided for the convenience of the general reader who is unfamiliar with the drugs.

Specific points:

1.     Line 47: is “midbrain and brainstem” OK?  Midbrain is a part of brainstem.  Please amend to this question.

2.     Line 68: “i.p” should be “i.p.”.

3.     Line 119: please use either “brain stem” or “brainstem” (line 47) throughout the text.

4.     Line 171: not “reduces” but “reduced”?

5.     Line 195: “1-methyl-1,2,3,4-..” should be “1-Methyl-1,2,3,4-..”.

6.     Line 199: please explain “5-HT3 receptor”.

7.     Line 200: it is not necessary to repeat the explanation of naloxone (see lines 189 and 190).

8.     Line 204: please specify what kind of neurotransmitter it is.

9.     Lines 209 and 232: please make a difference between 5-HT2A and 5-HT2A/2C to be clear.

10.  Lines 214 and 215: “hyperpolarization” should be “membrane hyperpolarization”.  “either pre- or post-synaptic” is difficult to understand.  Please amend this point.

11.  Lines 222 and 223: the explanation of epicatechin should be given in line 130 but not here.

12.  Line 235: “PL A2” should be “PLA2”.

13.  Line 247: it is not necessary to repeat the explanation of fluoxetine (see lines 44 and 45).

14.  Line 254: please write clearly what ketanserin is an antagonist of.

15.  Line 255: is the usage of “respectively” OK?

16.  Line 262: please put “membrane” before “depolarization”.

17.  Line 290 and others: is APPLI1” OK?  Not APPLI1” but APPL1”?  Please check this point.

18.  Line 291: it may be better to put “to be” before “drastically”.

19.  Line 303: “neurone” should be “neuron”.

20.  Line 311: please delete a space between period.

21.  Line 313: “- not” should be “-, not”.

22.  Line 327: “Courteix et al., 2013” cannot be found in [64].  Please amend this point.

23.  Line 331: “hargreaves” should be “Hargreaves”.

24.  Line 333: “DN4-interview” should be shortly explained.

25.  References: there are no authors in [44].  All of the references should be checked for correct citation.

26.  There may be more scientific or simple mistakes than those pointed out above.  This manuscript should be checked very carefully.

The quality of English language is good.

Author Response

Best regards

Reviewer 2 Report

The Authors should provide single section describing the serotonergic system with its 7 receptor subtypes, role, etc., and after that a subsection about alterations in neuropathic pani should be described.

In line with this, also a pain descending pathway and serotonin should be described. I suggest to present its also buy drawing a  figure.

The subsection 6 is written in a rather chaotic manner. The Authors mixed different drugs, while in the next pages they focused on serstoninergic receptors. This should be well organized.

When describe different drugs and their potent efficacy in neuropathic pain treatment (subs. 6) the Authors should clearly divide them. Also, for better presenting, please provide a table with a summary of the above mentioned

moderate English language corrections are required

Author Response

Best regards

Round 2

Reviewer 1 Report

This revised manuscript has been largely amended in response to my comments.  There are only minor points that should be considered, as follows:

1.     Line 80:         it is not necessary to repeatedly define “NOD” (see lines 57 and 58).

2.     Page 3: although Fig. 1 is added to this revised manuscript, there is no description about Fig. 1 in the text.  Please amend this point.

3.     Figure 1: please put “(PN)” following “Projection neurons” in the left-hand side figure.

4.     Line 135: what does “…” means?  Please make this point clear.

5.     Line 292: not “supports” but “support”?

6.     Lines 329-332: please put the definition of “i.m.” (see page 8)

7.     Line 350: not “use” but “used”?

The quality of English language is good.

Author Response

Best regards

Reviewer 2 Report

The paper is suitable for publication

English is fine

Author Response

No reply since the reviewer 2 judged the paper suitable for publication.

Best regards